# Prevalence and Characteristics of Violence against Paramedics in a Single Canadian Site

**DOI:** 10.3390/ijerph20176644

**Published:** 2023-08-24

**Authors:** Justin Mausz, Mandy Johnston, Dominique Arseneau-Bruneau, Alan M. Batt, Elizabeth A. Donnelly

**Affiliations:** 1Peel Regional Paramedic Services, Fernforest Division, 1600 Bovaird Drive East, Brampton, ON L6R 4R5, Canada; dominique.arseneau@peelregion.ca; 2Department of Family and Community Medicine, Temerty Faculty of Medicine, The University of Toronto, 500 University Avenue, Toronto, ON M5G 1V7, Canada; 3Peel Regional Paramedic Services, Tomken Division, 6825 Tomken Road, Mississauga, ON L5T 1N4, Canada; mandy.johnston@peelregion.ca; 4Faculty of Health Sciences, Queen’s University, 99 University Avenue, Kingston, ON K7L 3N6, Canada; alan.batt1@monash.edu; 5Department of Paramedicine, Monash University, Building H, 47-49 Moorooduc Hwy, Frankston, VIC 3199, Australia; 6School of Social Work, The University of Windsor, 167 Ferry Street, Windsor, ON N9A 0C5, Canada; donnelly@uwindsor.ca

**Keywords:** paramedics, emergency medical services, violence, occupational health and safety, mental health

## Abstract

Violence against paramedics has been described as a ‘serious public health problem’ but one that remains ‘vastly underreported’, owing to an organizational culture that stigmatizes reporting–hindering efforts at risk mitigation in addition to creating a gap in research. Leveraging a novel reporting process developed after extensive stakeholder consultation and embedded within the electronic patient care record, our objective was to provide a descriptive profile of violence against paramedics in a single paramedic service in Ontario, Canada. Between 1 February 2021 and 31 January 2023, a total of 374 paramedics in Peel Region (48% of the workforce) generated 941 violence reports, of which 40% documented physical (*n* = 364) or sexual (*n* = 19) assault. The violence was typically perpetrated by patients (78%) and primarily took place at the scene of the 9-1-1 call (47%); however, violent behavior frequently persisted or recurred while in transit to hospital and after arrival. Collectively, mental health, alcohol, or drug use were listed as contributing circumstances in 83% of the violence reports. In all, 81 paramedics were physically harmed because of an assault. On average, our data correspond to a paramedic filing a violence report every 18 h, being physically assaulted every 46 h, and injured every 9 days.

## 1. Introduction

Paramedics are an important element of Canada’s healthcare and public safety infrastructures, but their work exposes them to a myriad of health risks. In the years leading up to the COVID-19 pandemic, paramedics in Canada have been observed to have high rates of post-traumatic stress disorder (PTSD), depression, anxiety [1], chronic pain [2], disturbed sleep [3], and suicidality [4]—leading researchers and policymakers alike to describe the situation as a crisis in Canada [5]. Exposure to trauma compounds these risks [6,7,8,9,10,11], but one dimension of the crisis has remained understudied: the exposure of paramedics to violence.

Although situations that involve threats to physical safety are associated with an increased risk of adverse mental health among paramedics [12,13], including PTSD [7], the specific contribution of incidental or recurrent workplace violence remains largely unknown. When surveyed, a majority of paramedics in Canada [14] and elsewhere [15,16] indicate having experienced violence in the past year, and workplace injury statistics [17,18,19,20] point to violence as an occupational health risk with the potential for significant harm. Unfortunately, however, survey research often stops short of gathering granular, event-level data, and is also prone to recall bias. Likewise, injury statistics capture only the most serious incidents, leaving out less severe forms of violence such as verbal abuse, harassment, threats, or non-injurious assaults and underestimating the true prevalence as a result. Compounding the problem is a widely documented phenomenon of underreporting. For example, in a 2014 survey of paramedics from two Canadian provinces who reported exposure to violence, despite negatively affecting their well-being and job satisfaction, most did not report the incidents to their supervisors or police [14]. Researchers have since characterized violence against paramedics as a “serious public health problem” [16] (p. 11) but one that remains “vastly underreported” [21] (p. 494), despite creating the potential for significant physical [15,17,18,19,21] and psychological [22,23,24] harm. In an earlier study, we identified that the organizational culture within paramedicine may sustain underreporting by implicitly positioning the ability of paramedics to ‘brush off’ and ‘move on from’ acts of violence as an expected professional competency [25]. Within this construction, reporting itself can become stigmatized [25].

Underreporting is problematic for several reasons. First, underreporting leaves both researchers and paramedic service leadership on uncertain footing when developing risk mitigation strategies, creating the potential for policy interventions that may undermine patient safety. This was brought to light in 2012 during an inquest into the death of a Toronto man who—despite repeated calls to 9-1-1—died while paramedics ‘staged’ down the street from his apartment for 30 min because of unfounded scene safety concerns [26]. Second, existing research suggests that the path between potentially traumatic exposures and psychological sequelae can be mediated by prompt post-incident support-including debriefings, downtime, and attending to basic health and social needs [12,27,28]. Providing post-incident support depends on the paramedic service leadership being aware of the exposure—an unlikely scenario given the chronic underreporting of violence. Finally, underreporting creates an important gap in research. Having been framed as a ‘public health problem’ [21,29], it is incumbent upon researchers to generate basic epidemiological data on the prevalence, characteristics, risk factors, and health outcomes of violence as a threat to occupational health and safety—a process that requires robust data collection and risk surveillance.

Our team developed a novel reporting process [30] embedded within the electronic patient care report (ePCR) intended to overcome many of the organizational cultural barriers to reporting violence [25,31]. This new reporting process generates quantitative and qualitative data about violent encounters at the time of the event, as documented by the affected paramedic.

Therefore, as part of a larger program of research [32], our objective in the present study was to assess the prevalence of violence and describe the characteristics of violent encounters in a single paramedic service in Ontario, Canada.

## 2. Materials and Methods

### 2.1. Overview and Setting

This study is part of a larger research program, with a detailed description of the approach provided in an earlier publication [32]. For this study specifically, our methods involved a retrospective review of external violence incident reports (EVIRs) and ambulance call reports (ACRs) filed since the launch of the reporting process on 1 February 2021 through 31 January 2023. Our objectives were to measure the prevalence of violence against paramedics in a single paramedic service in Ontario, Canada, and to describe the characteristics of and circumstances that contribute to violent incidents.

This research is situated in the Regional Municipality of Peel. Peel Regional Paramedic Services (PRPS) is the sole provider of land ambulance paramedic service for the municipalities of Brampton, Mississauga, and Caledon, employing approximately 750 primary and advanced care paramedics (P/ACPs) and 60 paramedic supervisors who service a mixed urban/rural geography of 1200 km^2^ with a population of 1.38 million residents. The service responds to an average of 130,000 9-1-1 service calls per year, which—combined with its staffing and catchment area—makes the service the second largest in the province. The introduction of the violence reporting process occurred as part of a broader violence prevention program within the service that also included crisis intervention training, patient restraints, a public position of ‘zero tolerance’ for violence, and new policies that encouraged reporting.

### 2.2. Data Collection

A detailed accounting of the development of the EVIRs was described in an earlier publication from the research program [30]. The EVIR (see Appendix A) is a web-based form embedded within the electronic patient care record alongside the ACR and other incident reports. While the ACR collects administrative and clinical data about ambulance calls, the EVIR was built specifically to gather detailed information about violent encounters that occurred during a 9-1-1 call. The EVIR gathers quantitative and qualitative data about violent incidents using a combination of checkboxes and closed-ended questions about the type, location, source, and contributing circumstances of violence as well as questions about existing risk mitigation strategies (e.g., police presence and address flagging). The form also includes a free-text box where paramedics can type a detailed narrative description of the violent encounter with no character or word limit. Finally, the EVIR asks paramedics to indicate whether they were physically harmed or emotionally impacted (or both) at the time of reporting.

In addition to information about the violent incident itself, the form also automatically pulls key administrative data about the 9-1-1 call from the associated ACR. ACRs collect data on the patient’s presenting primary problem or injury and document the clinical care provided by the paramedics to facilitate continuity of care and quality assurance. When an EVIR is generated from a particular ACR, the call location code (e.g., residence, street, or hotel), dispatch priority (e.g., urgent or non-urgent), patient acuity level (as measured by the Canadian Triage Acuity Scale [33]), primary presenting problem code (e.g., shortness of breath or altered mental status), patient age and sex, and all dispatch event times are ‘pulled’ from the ACR and incorporated into the EVIR. In this respect, the EVIR compliments the ACR in capturing non-clinical data about paramedic service calls that resulted in a violent encounter with patients or members of the public.

Provincial documentation standards require paramedics to complete an ACR after every patient encounter and additionally require paramedics to complete an incident report in unusual circumstances that may have impacted service delivery, including threats to paramedic safety. During development, the EVIR was carefully vetted to ensure it is sufficiently detailed to stand on its own as an ‘incident report’. In addition to provincial standards, local policy also requires paramedics to complete an EVIR if they experience violence from the public—defined here as exposure to verbal abuse, threats, sexual harassment, assault, or sexual assault (see Table 1). When completing an ACR, paramedics are automatically prompted (by way of a ‘pop-up’ compliance rule) to complete an EVIR if they experienced violence during the 9-1-1 call. When an EVIR is filed, the expectation is that the form is reviewed by a paramedic supervisor within approximately 24 h of the event. On actioning a violence report, supervisors have the option to place a ‘hazard flag’ on the address of the 9-1-1 call if there is a risk of recurrent violent behavior from the perpetrator.

Our study window spanned a two-year period from the launch of the EVIR on 1 February 2021 through 31 January 2023. We included all EVIRs from paramedics who did not ‘opt-out’ of secondary use of the forms for research purposes in addition to the administrative data points described above from all ambulance call reports (regardless of violence) filed during the study period.

### 2.3. Analysis

We used descriptive and summary statistics to report on the prevalence of violence against paramedics in the service—defined here as the proportions of (1) unique 9-1-1 service calls that resulted in an EVIR being filed and (2) active-duty (i.e., not on leave) paramedics and supervisors who filed an EVIR during the study period. We also constructed a descriptive profile of the types, sources, and circumstances surrounding the violence the paramedics report.

In exploring contributing circumstances, we were specifically interested in whether response time to the scene and handover time in the receiving hospital were associated with an increased risk of violence. We assessed these risks in two ways: first, we plotted the intervals and assessed the data for normality using skewness and kurtosis tests. As expected, both response and handover time were positively skewed. To achieve a more normal distribution, we used percentiles and stakeholder consultation to ascertain a typical range of expected times and discarded unusually short (i.e., <10 min) and long (i.e., >180 min) intervals. The remaining data were subjected to Analysis of Variance (ANOVA) tests to assess group differences in the likelihood of (1) any violence and (2) assault when stratified by response and handover time. Second, local policy considers a response time over 15 min and a patient handover time longer than 30 min to be delayed. Accordingly, we dichotomized response and patient handover time and assessed the likelihood of violence using Chi-square tests above these thresholds.

All analyses were performed in SPSS version 28, and we followed the convention of accepting a *p*-value of less than 5% and confidence intervals that do not include the null value as indicating statistical significance.

## 3. Results

### 3.1. Prevalence of Violence

Between 1 February 2021 through 31 January 2023, 784 active-duty paramedics in the Region of Peel responded to 224,739 unique 9-1-1 service calls. In all, 374 paramedics filed a total of 941 EVIRs. This corresponds to a proportion of 0.4% of service calls that resulted in a documented incident of violence and 48% of active-duty paramedics reporting exposures to violence (Figure 1).

Among the paramedics reporting exposures to violence, the average number of reports filed was 2.51 (Standard Deviation [SD] 3.12, 95% Confidence Interval [CI] 2.19–2.82). Most (93%) paramedics who reported exposure to violence filed less than five reports during the study period. A total of 15 paramedics (4%) filed between 6 and 10 reports and 8 (2%) filed between 11 and 20 reports. Two paramedics filed more than 20 EVIRs.

### 3.2. Type Violence Reported

Detailed results are presented in Table 2. In declining order of frequency, the most common forms of violence reported were verbal abuse (*N* = 368 [40%]), assault (*N* = 170 [18%]), threats (*N* = 39 [4%]), sexual harassment (*N* = 20 [2%]), and sexual assault (*N* = 10 [1%]). However, paramedics completing an EVIR could select more than one type of violence, and many (*N* = 334 [36%]) reports documented multiple types of violence in a single incident. When dichotomized (assault/no assault), 40% of the reports (*N* = 379) involved some component of either physical or sexual assault alone or in combination with other forms of violence. Expressed as a rate, this corresponds to approximately one assault on a paramedic every 46 h.

### 3.3. Perpetrators and Locations of Violent Encounters

The majority (80%) of the violence reports listed the patient as the perpetrator, with family members (11%) and bystanders (2%) cited less frequently. Most (48%) violent incidents took place at the emergency scene; however, one third of the reports indicated multiple locations, either because the violent behavior recurred or persisted during transportation to hospital or after arrival.

Looking specifically at violence that occurred in the emergency department (ED) (*N* = 347 [36.9%]), 144 incidents (41%) involved either physical or sexual assault. Compared to all transfers to hospital, patients with a handover time >30 min were more likely to be (or become) violent (0.5% vs. 0.3%; Odds Ratio [OR] 1.76, 95% CI 1.54–2.00, *p* < 0.001). The risk was even more pronounced for physical or sexual assault where handover times exceeded the 30 min benchmark (0.2% vs. 0.1%; OR 2.38, 95% CI 1.96–2.93, *p* < 0.001). On average, paramedics waited 7 min longer to hand over the care of violent patients to ED staff (36.74 [95% CI 36.61–36.88] minutes vs. 44.31 [95% CI 41.86–46.76] minutes; f 53.41, *p* < 0.001) (Table 3).

Response time—either as a continuous variable or when examined categorically as ‘delayed’ (>15 min)—was not associated with an increased risk of violence.

### 3.4. Contributing Circumstances

In declining order of frequency, paramedics listed alcohol (25%), mental health (17%), cognitive impairment (8%), and drugs (6%) as contributing circumstances in violence reports. However, like the type of violence, the categories were often co-occurrent, with 22% of the reports listing more than one contributing circumstance. Mental health was listed as contributing in 35% of the reports either alone or in combination with other factors, and intoxication due to alcohol or drugs was similarly cited in 50% of the reports alone or in combination with other factors. Taken together, mental health or intoxication due to drugs or alcohol were listed as contributing circumstances in 85% of the reports.

### 3.5. Impact on Paramedics

A total of 211 paramedics indicated having been ‘emotionally impacted’ by the encounter at the time of the event—although current processes stop short of measuring diagnosable forms of psychological harm. In total, 81 paramedics (22% of those reporting violence and 10% of the overall workforce) were physically harmed because of a physical or sexual assault. This corresponds to a rate of one paramedic being physically injured from an assault every 9 days.

## 4. Discussion

Our objective was to estimate the prevalence with which paramedics in our study site experience violence in the course of their duties. We found that nearly half (48%) of the active-duty workforce reported exposure to violence during the study period, with a violence report being filed every 18 h—or, put differently, nearly one report for every 12-h paramedic shift. In all, 40% of the reports documented some form of physical or sexual assault, corresponding to a paramedic being assaulted every 46 h and physically harmed from an assault every 9 days. This prevalence of violence is concerning, and—to our knowledge—ours is the first study to gather such granular data at the time of the event, as documented by the affected paramedic.

Our findings shed new light on what has been described in research as a ‘serious public health problem’ [21] and have several important implications for research and policy. For paramedics in Canada, frequent exposure to violence compounds a myriad of existing workforce health issues that have almost certainly worsened since the COVID-19 pandemic. In our study site specifically, our earlier work found that one in four paramedics in Peel Region met the screening criteria for either PTSD, major depressive disorder, or generalized anxiety disorder as recently as February 2020 [34]. Situations that involve threats to physical safety increase the risk of adverse mental health outcomes, including PTSD [7], but in this population, violence as a threat to psychological health and well-being has not been widely studied. Where our findings contribute is in providing tangible data that quantifies the exposure to (what may be) a significant but understudied occupational health and safety risk. This opens the door for important epidemiological research on the potential dose–response relationship between exposure to violence and adverse mental health outcomes, such as PTSD.

Our reporting data stop short of gathering demographic information about the affected paramedics, and this, too, is an important area for future inquiry. Survey research, for example, suggests that the exposure to the amount, type, and severity of violence varies according to the age, gender, and career stage of the paramedic [15,21,29], but survey data are understandably vulnerable to recall bias. Linking our violence reporting process to routinely collected demographic information about the paramedic would allow for these and other hypotheses to be tested in a more robust way.

From a policy perspective, the frequency with which our paramedics were exposed to any form of violence and—importantly—of physical or sexual assault underscore the need for a comprehensive policy response. Employers have important responsibilities to protect the health, safety, and well-being of their staff that are enshrined in legislation, but the nature of paramedic work as a public safety profession makes enacting these responsibilities inherently challenging. With but a few exceptions, paramedics provide unscheduled and emergency health care at all hours of the day or night and in environments that are difficult—if not impossible—to control. In contrast to clinical settings where a patient seeking care may be (or become) violent, paramedics physically attend to the patient in their home or other location, placing them at a ‘tactical’ disadvantage and making them vulnerable to attack [35,36,37]. Identifying 9-1-1 calls that have a high risk of violence is critical in being able to organize a coordinated response that includes robust safety and security plans to reduce the risk of harm from violence—not just for the responding paramedics but also for the patients themselves as well as others at the scene. To this end, a recent visioning document commissioned by the Canadian Standards Association has called for the development of national standards and best practices related to violence prevention in paramedicine as a key priority over the next five years [38].

The relationship between the handover time at receiving hospitals and the risk of violence is especially concerning. Given the unique hazards of the out-of-hospital environment described above, we were surprised to see that 37% of the violence reports documented some form of violence occurring after arrival at a hospital emergency department. Where handover times exceeded the 30 min benchmark, the risk of assault more than doubled. These findings underscore the need for improved coordination with our hospital colleagues, given the risk of harm to paramedics, hospital staff and visitors, and—not least—patients.

Finally, our findings raise important questions about how society should respond to acts of violence perpetrated against paramedics and other healthcare providers or first responders. Although the paramedic context may be more amenable to measurement, the issue of violence in healthcare settings is in no way unique to paramedicine [39,40]. There have been calls from professional associations representing physicians [41], nurses [42], and other healthcare professionals to address growing reports of hostility toward healthcare providers during the COVID-19 pandemic [43]. In response, the Canadian government introduced legislation to make intimidating a healthcare worker a criminal offense [44]. There is also a new bill currently in the House of Commons that, if passed, would amend the Criminal Code of Canada to impose harsher sentences on people who assault healthcare professionals or first responders [45]. Violence against healthcare providers is a complicated issue, not least because violent acts may be perpetrated by people in times of crisis without the criminal intent (or the capacity to form the intent) to harm the victim. Although we agree that criminal assault should be appropriately prosecuted through the justice system, the degree to which the violence perpetrated against healthcare providers—including the incidents described here—meets the threshold of criminality is, again, unknown.

### Limitations

Our findings should be interpreted within the context of certain limitations. Although we have invested considerable effort to encourage reporting, we recognize that we are operating within an organizational culture that may still consider violence ‘part of the job’ [25]. We are aware anecdotally of several incidents (including injurious assaults) that were not documented via the new reporting process. We also have no means of cross-referencing violence reports with injury reports. This means that our prevalence estimates are likely conservative, particularly for forms of violence that the paramedics may consider less severe, such as verbal abuse or threats. Second, and as noted above, our reports are decoupled from several other streams of data that are important in understanding the scope of the problem, particularly its potential for harm. Linking violence reports to employee demographics, injury reports, and workplace insurance claims (i.e., for lost time from work) is an important line of inquiry to advance in future research. This will be especially important for understanding the potential long-term effects of incidental or cumulative exposure to violence. Finally, we have taken the reports filed by the paramedics at face value, without attempting to ascertain the veracity of the paramedics’ allegations of assault (for example), although we have no reason to doubt them.

## 5. Conclusions

Over a two-year period in our study, we found that, on average, a member of the paramedic service filed a violence report every 18 h, was physically or sexually assaulted every 46 h, and was injured because of an assault every 9 days, with 48% of the active-duty workforce reporting exposure to violence. Our novel, point-of-event reporting process opens the door for robust epidemiological research on violence as a risk to occupational health, safety, and well-being and emphasizes the need for robust policy and training to strengthen both paramedic and community safety.

## Figures and Tables

**Figure 1 ijerph-20-06644-f001:**
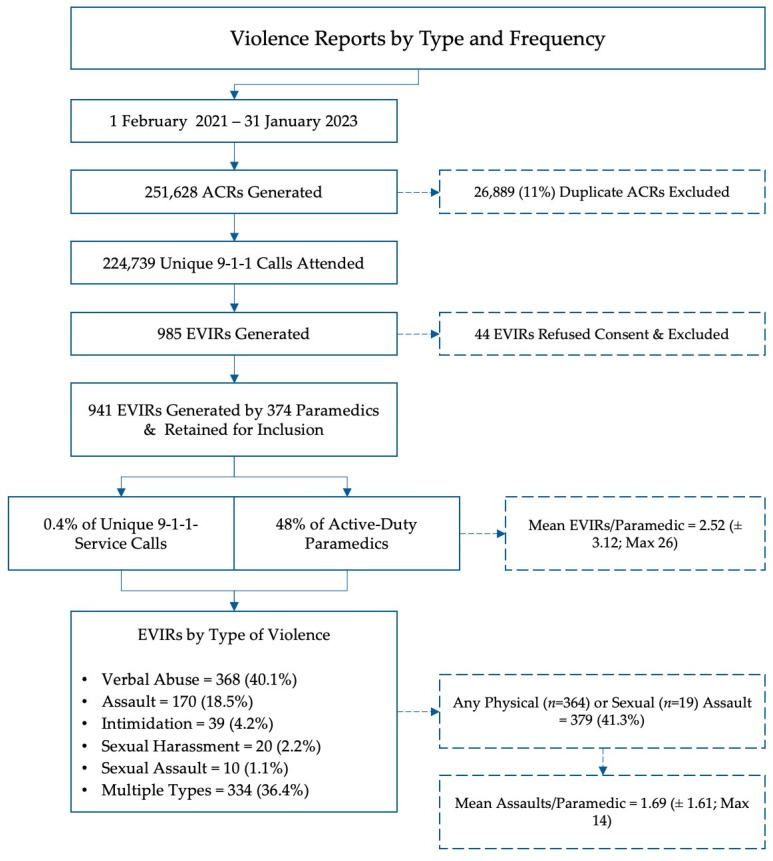
Process flow of included and excluded cases.

**Table 1 ijerph-20-06644-t001:** Definitions of violence included in the External Violence Incident Report (EVIR), adapted from Bigham et al. (2014).

Type of Violence	Definition
Verbal Abuse	Offensive or hateful language, yelling, or screaming with the intent of offending or frightening the paramedic.
Intimidation	Purposely threatening, following, or using gestures to offend or threaten the paramedic.
Sexual Harassment	Sexual propositioning or unwelcome sexual attention from a perpetrator. Humiliation or offensive jokes or remarks with sexual overtones, suggestive looks, or physical gestures.
Assault	Physical attack or attempt to attack, for example through punching, kicking, or using a weapon with the intent of causing bodily harm.
Sexual Assault	Indecent assault, for example, brushing, touching, or groping the genitals or breast area.

**Table 2 ijerph-20-06644-t002:** Detailed breakdown of EVIR characteristics.

Parameter	Missing	Count	%
Type of Violence Reported	0		
Verbal Abuse		368	39.1%
Assault		170	18.1%
Intimidation		39	4.2%
Sexual Harassment		20	2.1%
Sexual Assault		10	1.1%
More Than One Type		334	35.5%
Assault/No Assault	0		
Any Physical Assault		364	38.7%
Any Sexual Assault		19	2%
Any Physical or Sexual Assault		379	40.3%
Source of Violence	0		
Patient		736	78.2%
Family Member (of patient)		103	10.9%
Other Person		31	3.3%
Bystander		25	2.7%
More Than One Source		46	4.9%
Location of Violent Incident	2		
At Scene		447	47.5%
In Transit		79	8.4%
At Hospital		107	11.4%
More Than One Location		306	21.7%
Any Violence at Hospital		347	36.9%
Contributing Circumstances	0		
Alcohol		226	24%
Mental Health		152	16.2%
Cognitive Impairment		71	7.5%
Drugs		51	5.4%
None of the Above		237	25.2%
More Than One Contributor		204	21.7%
Any Alcohol or Drugs		461	49%
Any Mental Health		317	33.7%
Any Mental Health or Substance Use		778	82.7%
Outcomes	0		
Physically Harmed		81	8.6
Emotionally Impacted		211	22.4

**Table 3 ijerph-20-06644-t003:** Mean handover times for violent/non-violent patients in receiving emergency departments.

Parameter	*N*	Mean	SD	Median	IQR	Skewness	Kurtosis	F	*p*
Any Violence	653	44.31	31.87	32.46	31.54	1.77	3.09	53.41	<0.001
No Violence	152,006	36.74	23.36	29.00	23.08	2.29	6.43		
Any Assault	293	45.30	32.10	34.00	30.99	1.76	3.06	30.66	<0.001
No Assault	152,366	36.76	26.76	29.00	23.13	2.29	6.42		

## Data Availability

Data for this study may be shared with interested researchers on a case-by-case basis, subject to a privacy review and formal data sharing agreement.

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
