# Peer review of "Prevalence and Characteristics of Violence against Paramedics in a Single Canadian Site"

_ijerph, 2023, doi:10.3390/ijerph20176644_

Round 1

Reviewer 1 Report

I appreciate the opportunity to read and review the paper entitled: “Prevalence and characteristics of violence against paramedics in a single Canadian site”. The research tries to provide a descriptive profile of violence against paramedics in a single paramedic service in Ontario, Canada. Hopefully you find my comments helpful. 

·       The introduction starts with a general statement about paramedics in Canada facing health risks and high rates of mental health issues. However, it would be beneficial to provide more specific information on the significance of the study and why understanding violence exposure among paramedics is important. Clearly state the gap in the current literature and the potential implications of the research findings.

·       The introduction mentions the novel reporting process developed by the team, but it does not clearly state the specific objectives of the present study. Be explicit about the main research questions or objectives that this study aims to address, such as assessing the prevalence of violence exposure and understanding the characteristics of violent encounters among paramedics.

·       In the method section, it would be helpful to explicitly state the primary purpose of the EVIR (Electronic Violence Incident Report) and how it complements the ACR (Ambulance Call Report). While it briefly mentions that the EVIR is built specifically to document violent encounters, it should be clarified how the EVIR serves as a supplementary report to the ACR. Readers may better understand the rationale behind using both forms and how they complement each other in capturing relevant data about violent incidents.

·       The paper mentions that the study window spans a two-year period from February 1, 2021, through January 31, 2023. It would be beneficial to provide some justification or reasoning for choosing this specific time frame. Explaining why this period is relevant and appropriate for the study could enhance the overall validity and credibility of the research.

Overall, the paper has significant strengths and valuable contributions to the literature. Implementing these recommendations will enhance its quality and readiness for publication.

Author Response

Dear colleague,

Thank you for your very helpful review of our paper. We have incorporated your recommended edits into our revised manuscript. Please refer to the attached file for more specific details.

Thank you kindly,

Justin

Reviewer 2 Report

The manuscript is well written, straight-forward, and provides evidence of workplace violence issues associated with Canadian paramedicine workers responding to emergencies.  The manuscript is significant in its findings and its use of a special reporting form.

Minor edits include (the minor issues could have been due to the generation of the pdf from its original source):

Line 77 - why is "objective" bolded?

Line 102 - "in" should be eliminated from the sentence so it reads: "publication from a research program"

Figure 1 - the mean EVIRs/Paramedic should be reported as 2.52 not 2.51.   941/374 = 2.516 (which should be rounded up to 2.52).

Table 2.  Under PARAMETER, the first line reads:  "ype of Violence Reported".  Obviously it should be "Type of Violence Reported".

Table 2.  Under the PARAMETER "Assault/No Assault" is the line under "Assault/No Assault" needed?  Other categories below do not have a line under a category heading.

In the Supplemental attachment "External Violence Incidence Report", it cites a "hazard tag" being created or extended.  This hazard tag was never mentioned in the manuscript, including its function.  Perhaps a sentence or two can be added to the manuscript describing what it is and its function.

Author Response

(The authors gave the same response as above.)
